# The Disputed Middle Ground: Tibetan Mādhyamikas on How to Interpret Nāgārjuna and Candrakīrti

**John Powers**

School of Humanities and Social Sciences, Deakin University, Waurn Ponds, VIC 3216, Australia; john.powers@deakin.edu.au

**Abstract:** By the twelfth century, a broad consensus had developed among Tibetan Buddhists: The Middle Way School (Madhyamaka) of Nāgārjuna (c. 2nd century), as interpreted by Candrakīrti (c. 600–650), would be normative in Tibet. However, Tibetans had inherited various trajectories of commentary on Madhyamaka, and schools of thought developed, each with a particular reading. This article will examine some of the major competing philosophical stances, focusing on three figures who represent particularly compelling interpretations, but whose understandings of Madhyamaka are profoundly divergent: Daktsang Sherap Rinchen (1405–1477), Wangchuk Dorjé, the 9th Karmapa (1556–1603), and Purchok Ngawang Jampa (1682–1762). The former two contend that Nāgārjuna's statement "I have no thesis" (*nāsti ca mama pratijñā*) means exactly what it says, while the latter advocates what could be termed an "anthropological" approach: Mādhyamikas, when speaking as Mādhyamikas, only report what "the world" says, without taking any stance of their own; but their understanding of Buddhism is based on insight gained through intensive meditation training. This article will focus on how these three philosophers figure in the history of Tibetan Madhyamaka exegesis and how their respective readings of Indic texts incorporate elements of previous work while moving interpretation in new directions.

**Keywords:** Buddhist philosophy; Madhyamaka; epistemology; Nāgārjuna; Candrakīrti; Daktsang Sherap Rinchen; Wangchuk Dorjé; Purchok Ngawang Jampa

## 1. Introduction

When Tibetans began importing Buddhism in the seventh century, they inherited a philosophically rich corpus of thousands of written works, along with oral commentarial traditions from various Indian schools. This was supplemented (and often challenged) by material that came from China and was accompanied by Buddhist masters seeking to disseminate their views and gain adherents. Buddhism became the state creed and increasing numbers of Tibetans received ordination and began to engage in intensive study, often with Indian or Chinese masters who traveled to the Tibetan Plateau to spread the Dharma.

Historical sources report that there was vigorous debate regarding the relative status of imported philosophical and practice traditions. According to several accounts, these matters were settled at the conclusion a synod in the ninth century held at Samyé (bSam yas) Monastery, during which a Chinese faction led by the Chan 禪 master Heshang Moheyan 和尚摩诃衍 (Tib. Hwa shang Ma ha ya na, fl. 8th century) and Indian monks headed by Kamalaśīla (fl. 8th century) propounded, respectively, a subitist vision of the Buddhist path and a gradualist paradigm. The former claimed that buddhahood can be attained all at once, in a flash of awakening, while the Indian cohort defended a traditional model according to which the path to buddhahood proceeds in stages.[1] At the end of the debate, King Tri Songdetsen (Khri Srong lde'u btsan, r. 754–c. 799) declared that the Indian side was victorious, Chinese Buddhism was characterized as heretical, and Nāgārjuna's philosophical system would henceforth be enshrined as the supreme articulation of Buddhist

principles.[2] The real situation was probably more messy and ambiguous than what the descriptions of straightforward doctrinal differences and a decisive outcome found in Tibetan histories—written several centuries later by Buddhist clerics intent on glorifying Buddhism and aiding its dissemination—depict, but this narrative became the dominant one among Tibetan Buddhists.[3] This is still the case today.

## 2. The Riddles of the Middle

Despite the designation of Nāgārjuna's system as the state ideology, numerous questions remained regarding how it should be interpreted and exactly what the Master and his authoritative commentator Candrakīrti understood to be Buddhism's "Middle Way." Along with a substantial corpus of Indic works that were translated into Tibetan, Mādhyamikas in the Land of Snows inherited trajectories of exegesis that contained incompatible readings. These were further elaborated as Tibetans debated the conceptual ramifications of Indian sources. A rich tradition of oral dialectical debate that drew much of its material from Indic sources developed in Tibet. This format is most closely associated with the Geluk (dGe lugs) order founded by Tsongkhapa Losang Drakpa (Tsong kha pa bLo bzang grags pa, 1357–1419), but is also practiced by other orders, including the Kagyü (bKa' brgyud), Sakya (Sa skya), and Nyingma (rNying ma).

Some of the most heated controversies centered on ostensibly antinomian statements by Nāgārjuna and Candrakīrti (as well as other luminaries, including Āryadeva (c. 163–261) and Buddhapālita (n.d.) (c. 500–560), such as Nāgārjuna's assertion:

> If I had any thesis,
>
> That error would apply to me.[4]
>
> But I have no thesis,
>
> And so I do not have this fault.
>
> If I were to apprehend anything
>
> By means of perception and the other epistemic instruments,
>
> Then I would engage in affirmation or rejection; but
>
> Because I do not do so, no such charge can be leveled against me.[5]

Exactly what Nāgārjuna meant by "no thesis" has been differently understood by various commentators. Some, as we will see, took him at his word and claimed that Mādhyamikas pursue a purely negational approach, drawing out the conceptual implications and unwanted consequences of opponents' positions through *reductio ad absurdum* (*prasaṅga*; Tib. *thal 'gyur*) analyses, while not putting forward any positive theses of their own. According to others, Madhyamaka *is* a position, but one that cannot be put into words because verbal concepts are incapable of accurately conveying the Dharma as understood by buddhas. Still other Mādhyamikas view words and concepts as dangerously inclined toward distortion, but add that they are all we have for communicating ideas and discussing how to interpret Buddhist teachings. This leads to further questions: Is a thesis a claim regarding ultimate reality? Must a thesis be expressed verbally, or can it be something intuitively understood? Do all theses involve positive assertions?

These issues correlate with tensions in both Nāgārjuna's and Candrakīrti's treatises.[6] Candrakīrti discusses epistemic warrants at length; in some places, he emphasizes the deceptiveness of conventional truths, while in others he endorses the use of conventional epistemic warrants in accordance with how "the world" (*loka*; Tib. *'jig rten*) employs them to arrive at knowledge in which people can have (at least provisional) confidence. Tibetan exegetes tend to emphasize one or the other side of this tension. The debates discussed in this article hinge on relative levels of emphasis: broadly speaking, Gelukpa readings highlight implications of conventional truth (*saṃvṛti-satya*; Tib. *kun rdzob bden pa*; literally, "obscuring/deceiving truth") as convention—that is, what is accepted in common discourse—while their Sakya and Kagyü opponents stress the notion that it is deceptive

and false.[7] An example of Candrakīrti's analysis of epistemic warrants is the statement in *Commentary on Four Hundred Verses*:

> Therefore, to ascribe the status of perception to sensory cognitions and to think that those cognitions function as epistemic warrants for their objects is utterly indefensible. From a mundane perspective, an epistemic warrant is regarded as a nondeceptive cognition. The Blessed One taught that cognition is a conditioned phenomenon, and therefore is false and deceptive, just like an illusion. Being false, deceptive, and illusory, it cannot be nondeceptive, because things appear to it in a way that is different from the way they actually are. Therefore, it is not reasonable to regard such a cognition as an epistemic warrant because then all cognitions would end up being epistemic warrants.[8]

There is general agreement among Madhyamaka exegetes that something's being deceptive entails that it does not exist in the way that it appears. Conventional truths are perceived by ordinary beings as having inherent nature (*svabhāva*; Tib. *rang bzhin*), but in fact they are composites of particles that are produced by causes and conditions external to themselves; they change in every moment; and they pass away due to causes and conditions. Thus, their mode of existence and mode of appearance are discordant. A core issue in discussions of conventional truth is whether it is still possible to validly recognize regularities of cause and effect and arrive at reliable knowledge that produces predictably repeatable results.

Early Tibetan philosophers inherited the questions raised by their Indian predecessors. They focused on many of the same issues, and they also developed new trajectories of commentary. Some, including Chaba Chökyi Sengé (Phya pa Chos kyi seng ge, 1109–1169), viewed Candrakīrti's Madhyamaka as a deviation from Nāgārjuna's intent. Chaba characterized it as nihilism that would leave those foolish enough to accept it unable to function in the world. Others, for example Batsap Nyima Drakpa (Pa tshab Nyi ma grags pa, b. 1055), took Candrakīrti at his word and read him as rejecting any appeal to validity in epistemic instruments. A third position is represented by Mapja Jangchub Tsöndrü (rMa bya Byang chub brtson 'grus, d. 1185), who believed that Candrakīrti's writings support a robust account of knowledge acquisition.

Batsap claims to follow Nāgārjuna's lead as expressed in *Reply to Objections* (*Vigrahavyāvartanī*): "I do not accept any epistemic instruments because such instruments and the objects evaluated by them are both refuted in *Reply to Objections*. If there were epistemic instruments, there would have to be objects to be evaluated, and such objects do not exist."[9] He explains that the results of application of one epistemic instrument cannot function on their own: they are part of a network of justification. Any attempt to develop foundational practices within the domain of conventional truth is doomed to fail because the objects for which they seek reliable knowledge are of a merely apparent nature. The Madhyamaka doctrine of universal emptiness precludes any recourse to epistemic instruments; Mādhyamikas do not attempt to chart the way things are, but rather employ reasoning to deconstruct the misconceptions of others.

Chaba presents a more positive position regarding a Madhyamaka view. He contends that it is absurd to assert that Mādhyamikas could effectively argue against wrong views if they lacked an understanding of what is correct. The claim that all phenomena are empty of inherent existence is a statement about reality: "You must be claiming that being empty of ultimate entities is the basic condition of cognizable things, and so you cannot maintain that you have no thesis."[10] It would be empty nihilism to refute opponents' positions without grounding in a correct understanding of the nature of reality. Mādhyamikas employ perception and inference to understand the character of "true objects" (*yul bden pa*), which is what is comprehended by the omniscient minds of buddhas.

Mapja agrees with Chaba that Mādhyamikas must have a view and that a denial of this would result in philosophical incoherence: "If you have no position of your own, then there can be no position of others either. If that is the case, then what it is that you wear

yourselves out refuting?"[11] For a Mādhyamika (or an adherent of any other system, for that matter), it is necessary to have a sense of what is correct. But Mapja also thinks that even conventionally speaking this cannot be grounded on any objective facts, because there are no such facts. He rejects the idea (which he associates with Svātantrika) that epistemic instruments are able to correctly discern particulars (*svalakṣaṇa*; Tib. *rang gi mtshan nyid*):

> Things like arising and cessation are like dreams and illusions. They are merely appearances that occur in a deluded mind. The sort of particulars in which the Svātantrikas believe do not exist even in terms of the conventional truth.[12]

In a sense Mapja splits the difference between Chaba and Batsap: Mādhyamikas have views, and they can employ epistemic instruments conventionally, but there is no objective reality that accords with a foundationalist epistemology. Epistemic instruments cannot validly discern particulars, and so there is no reliable inference "through the power of the object" (*dngos po'i stobs zhugs*). Consciousness would have to be able to apprehend particulars in order for the sort of knowledge sought by Svātantrikas to be possible, but it does not. According to Mapja, the omniscience of buddhas is connected with a final cessation of mind (*sems*; Skt. *citta*) and mental states (*sems 'byung*; Skt. *caitta*) that results from moving beyond any sort of foundational epistemology and realizing things as they really are.

These three attempts to reconcile tensions implicit in Nāgārjuna's thought highlight some of the trajectories of commentary among Tibetans who identified as Mādhyamikas. Their conflicting readings continue to resonate in Tibetan intellectual circles, and they constituted the philosophical basis within which the authors we will now consider—Daktsang, Wangchuk Dorjé, and Purchok—developed their exegeses. Contemporary scholars, both Asian and Western, still wrestle with inherent ambiguities and ellipses in the works of Nāgārjuna and his followers. Can Madhyamaka provide a robust account of the conventional sufficient to warrant knowledge and effective action? Is it a purely negative tradition, a parasitical approach to philosophy that only exists as a critique of others' systems? Is it mysticism, a retreat from conceptual thought into a quietist stance based on intuitive realization of ultimate reality? Interpreters basing themselves on the Madhyamaka corpus have proposed readings along all of those lines, as well as various amalgamations of possible expositions.

### 3. Daktsang's Critique

One of the most influential Tibetan contributions to Madhyamaka debates was Daktsang's doxographical treatise *Freedom from Extremes Accomplished through Comprehensive Knowledge of Philosophy*.[13] Daktsang, following a model employed by other Tibetan exegetes, ranked Buddhist tenet systems hierarchically, with Prāsaṅgika Madhyamaka at the apex, followed by, in descending order, Svātantrika Madhyamaka, Cittamātra, Sautrāntika, and Vaibhāṣika. In the "Madhyamaka" chapter, Daktsang trenchantly critiques Tsongkhapa's system, charging him with "eighteen great burdens of contradiction" (*'gal khur chen po bco brgyad*) in his presentation of the two truths. Daktsang's discussion is wide-ranging and encompasses a vast corpus of literature, but four main points stand out: (1) his discussion of how to understand the implications of deceptiveness and falsity in relation to conventional truths; (2) his presentation of the nature of buddhahood, particularly his contention that buddhas only perceive ultimate truths and so do not share the perceptions of ordinary beings, who operate on the conventional level; (3) how Mādhyamikas should employ *reductio* arguments; and (4) his characterization of epistemology as a merely mundane science, on a par with such fields of knowledge as medicine and grammar, and thus not a uniquely Buddhist subject.

Daktsang characterizes Prāsaṅgika Madhyamaka as thoroughly antifoundational. Mādhyamikas, *qua* Mādhyamikas, make no assertions and only engage in debate with opponents by deconstructing their theses by means of *reductio* analyses. They put forward no positive theses of their own, and their philosophical work is complete when opponents realize the flaws of their positions and abandon them.

According to Daktsang, Tsongkhapa's system lacks the rigor of true Prāsaṅgika because he attempts to merge the foundationalist Sautrāntika school of Dignāga (c. 480–540) and Dharmakīrti (c. 7th century) with Madhyamaka, and the outcome is an incoherent muddle of mutually incompatible philosophical approaches. Madhyamaka understands that conventional reality is inflected with error, and all perceptions of ordinary beings are colored by false imputation of inherent nature. This causes them to misunderstand their sensory inputs and mental impressions prompted by them. Mundane perceptions are deceptive (*bslu ba*; Skt. *visaṃvādaka*) and mistaken regarding the nature of their objects; in light of this, it is nonsensical to distinguish "true" and "false" falsities, as Tsongkhapa attempts to do. Daktsang sums up his position: "Being in error about an object contradicts being an epistemic warrant for it."[14]

Tsongkhapa believes that it is possible to have nondeceptive knowledge of conventional reality and to have confidence in the epistemic warrants we use to arrive at it. Following Dharmakīrti, Tsongkhapa contends that in spite of their deceptive appearance conventionalities can be understood correctly through veridical perception (*mngon sum*; Skt. *pratyakṣa*)—that is, data that is not disconfirmed by subsequent input—and correct inference (*rjes su dpag pa*; Skt. *anumāna*), which is founded on valid perception.

According to Tsongkhapa's interpretation of ultimate truth (*don dam bden pa*; Skt. *paramārtha-satya*), it should be understood as a nonimplicative negation (*med dgag*; Skt. *prasajya-pratiṣedha*) that does not imply any positive phenomena in place of that which is negated. He further contends that the ultimate truth can be verified through reasoning and directly perceived by noble beings (*'phags pa*; Skt. *ārya*)—that is, those who have attained the path of seeing (*mthong lam*; Skt. *darśana-mārga*) and higher levels.

Daktsang rejects these ideas. Ordinary perception is conditioned by ignorance and is deceived by the way things appear to it. Buddhas' cognitions, however, are entirely free from error. Buddhas only perceive reality as it is, viz., as ultimate truth. Both Tsongkhapa and Daktsang agree that buddhas are omniscient, but exactly what this means is understood differently: Daktsang asserts that the purview of their awareness is untinged by error, and their perceptions are free from the dichotomies of subject and object, existence and nonexistence, and other extremes that result from ignorance. Conventional truth encodes all of these factors, and so Daktsang concludes that buddhas do not engage with it. For these reasons, discussions of truth and warrant only operate in the realm of mundane transactions and so have no place in Madhyamaka properly understood.

The key to understanding Daktsang's interpretation lies in his distinction of three distinct contexts:

> I have understood that in general all teachings of the Victor—and in particular the scriptures of Nāgārjuna and his heirs—can be put into practice with great ease if one relates their statements to three contexts: (1) that of no examination and analysis (*ma brtag ma dpyad pa*); (2) that of slight analysis (*cung zad dpyad pa*) on the basis of rational cognition; and (3) that of thorough analysis (*legs par dpyad pa*) on the basis of the ineffable.[15]

The first is the epistemic mode of ordinary people unconcerned with the questions that engage philosophers. They employ epistemic instruments, including perception, inference, verbal testimony (*lung*; Skt. *śabda*), and analogy (*nye bar 'jal ba*; Skt. *upamāna*), to make sense of their surroundings and to make decisions. As Candrakīrti describes this situation, "What the six unimpaired senses apprehend in the mundane world is held to be real by the world. The rest, according to the world, is deceptive."[16] Daktsang's approach is anthropological: Mādhyamikas describe mundane epistemic practices but make no commitments regarding their ultimate validity. And Mādhyamikas do not assert that such judgements actually describe the world as it is; or even that there *is* a way the world is.

The second context applies the critique of emptiness to the phenomena of experience and demonstrates that they are dependent arisings, and so they lack inherent existence. For those operating in this realm, only insight into ultimate reality has the status of an epistemic instrument. In the second context, everything is understood to be merely conventionally

true, deceptive, and overlaid with false impressions, and one comprehends emptiness as the ultimate truth.[17]

The third context is the purview of noble beings: they only perceive ultimate reality, and no words or concepts can convey any sense of what it is like to operate within this perspective. Their cognitive world is indescribable and inconceivable; even emptiness and the distinction drawn between the two truths in the second context are no longer operative because they are merely appearances. Things are not even dependently arisen, "emptiness" is a mere term, and there is no possibility of a valid epistemic instrument. For such beings, the ultimate reality is a "disclosed content" (*rnyed don*) apprehended from "the rational perspective of noble beings free from error."[18]

Tsongkhapa, however, wants to retain validity for mundane epistemic practices and a grounding for ethics. He also views this as essential for the core Buddhist doctrine of dependent arising (*rten cing 'brel bar 'byung ba*; Skt. *pratītya-samutpāda*) to make sense. By means of valid epistemic instruments, we can know that things arise dependently, and that karma operates as described in Buddhist sources. Persons and events only exist conventionally, but within that domain is it possible for them to be real, and for us to know them as real. If this is the case, then there can be valid epistemic warrants that enable us to acquire knowledge, as well as corroboration for the claims we make about the operations of the things of our worlds.

Daktsang rejects all of this: Prāsaṅgika Madhyamaka, on his understanding, is a radical antirealism. Epistemic instruments only function in the first context, that of the unexamined and unanalyzed perspective. The second context, within which Madhyamaka operates, is characterized by understanding of the falsity of conventions once they are subjected to the critique of emptiness. This leads to comprehension of ultimate truth. But Daktsang cautions that in the realm of ultimate analysis there is no such thing as ultimate reality—or, for that matter, absence of ultimate reality—because there can be no epistemic instruments capable of delivering such knowledge. Reality understood in this domain is not an object—it transcends cognition and cannot accurately be characterized either positively or negatively. Even describing things as empty is only valid within the second context; in the third, all words cease and only nonconceptual experience remains.

> Finally, for the most part, the explicit teaching of the Mother of the Victors[19] completely denies that things exist, that they do not exist; that they are permanent or impermanent; or that they are empty or nonempty; and that there is anything to be apprehended in any way at all ... Now, consider the absence of fabrications that is free from the four extremes even conventionally:[20] it cannot be posited even from the perspective of slight analysis, let alone from the erroneous perspective of someone who carries out no inquiry or analysis. The reason is that this is conveyed in the inexpressible and inconceivable perspective of someone who has undertaken a thorough analysis of all the extremes of existence and nonexistence.[21]

Daktsang's attack on Tsongkhapa's understanding of Madhyamaka can be summarized in four principles: (1) epistemic warrants entail foundationalism, and Prāsaṅgika is antifoundational; (2) any claim or thesis can be deconstructed by means of Madhyamaka analysis, and so all expressions of knowledge can be exposed as false; (3) privileging any set of faculties as warranting is arbitrary and cannot be justified by anything outside self-contained epistemic systems; and (4) knowledge requires that the knower be correct about the object of knowledge, and we are always mistaken in some respect about any object.

Daktsang was operating within a shifting political climate. The Sakya school with which he was affiliated had been the power elite in central Tibet during the hegemony of the Mongol Empire (1206–1368), during which Sakyapa hierarchs served as the Mongols' regents in Tibet. The end of Mongol control significantly diminished the power and political reach of the Sakyapas. The nascent Gelukpa order challenged both their political

position and their philosophical system. The Mongols remained a potent force throughout Asia, however, and in Tibet religious groups sought the patronage and military backing of Mongol leaders. The Gelukpas were engaged in intermittent armed conflicts with rivals, particularly the Kagyüpas, and the Sakyapas also continued to press their claims to control in central Tibet.

During Tsongkhapa's time, the Gelukpas had avoided entanglement in political conflicts and had gained a reputation for strict adherence to the rules of monastic discipline and excellence in scholarship. As their power and influence grew, however, other orders came to view them as a threat and attacked them, both philosophically and militarily. In 1498 control of the Great Prayer Festival (sMon lam chen mo) was wrested from Gelukpa control, and during the sixteenth century the kings of Tsang (gTsang), who were patrons of the Kagyüpas, actively suppressed the Gelukpas. In 1642, however, the fifth Dalai Lama, Ngawang Losang Gyatso (Ngag dbang blo bzang rgya mtsho, 1653–1703), was installed as Tibet's most powerful figure with the help of Mongol armies, and several monasteries that had been seized by the Kagyüpas were returned to Geluk control. The Gelukpas refrained from a wholesale pogrom against their former adversaries, but their ascent saw a reduction in power and influence among the other orders.

**4. The Gelukpa Response**

Daktsang's critique of Tsongkhapa's Madhyamaka potentially undermined the entire Gelukpa project, and this was understood by leaders of the order. The fifth Dalai Lama called on his compatriots to defend their order's founder and his system.[22] The first to respond was Losang Chökyi Gyeltsen (bLo bzang chos kyi rgyal mtshan, 1567–1662), the fourth Panchen Lama, who characterizes Daktsang's presentation of Madhyamaka as dangerous nihilism.[23] Ignoring Daktsang's claim that he adopted a Madhyamaka *prasaṅga* approach and asserted no theses of his own, the Panchen Lama employs a dialectical debate style and accuses his opponent of endorsing the opposite of every "contradiction" that he attributes to Tsongkhapa. This includes positions Daktsang does not affirm and some that he explicitly rejects. Much of the critique is well argued and represents a serious response to Daktsang, but it is flawed by these factors.

The second Gelukpa response, by Jamyang Shepa ('Jam dbyangs bzhed pa'i rdo rje Ngag dbang brtson 'grus, 1648–1721/2), is less philosophically satisfying.[24] It mainly relies on invective directed toward Daktsang, hyperbolic sarcasm, and ad hominem attacks. Jam-yang Shepa repeats most of the Panchen Lama's points and apparently believes that the matter has already been settled. His task is to heap abuse on Daktsang for his temerity in attacking Tsongkhapa, who is regarded in Geluk tradition as an emanation of Mañjuśrī, the bodhisattva of wisdom.[25]

All three of the Gelukpas who composed responses to Daktsang's critique (the third being Purchok) also employ a further polemical device: they refer to a document that only appears in Geluk-produced collections of Daktsang's works, a verse paean to Tsongkhapa that purports to be a repentance written late in life after Daktsang realized the error of his youthful philosophical indiscretions.[26] The author refers to Tsongkhapa as an emanation of Mañjuśrī, proclaims that his Madhyamaka is faultless and beyond any possible reproach, and vows to worship Tsongkhapa for the remainder his present life and in all future lives.

There are a number of problems with this text. Firstly, there are several extant similar works in which members of other orders who published trenchant critiques of Tsongkhapa repent and declare their unfailing devotion.[27] It is not mentioned by anyone outside Geluk circles and is not found in non-Geluk editions of Daktsang's works. Moreover, *Comprehensive Knowledge of Philosophy* was written toward the end of Daktsang's life, and so presumably represents his mature thought on Buddhist philosophy. Finally, the paean provides no indication of exactly what aspects of Daktsang's critique were later realized to be erroneous or what insights from Tsongkhapa led to his conversion.

## 5. Wangchuk Dorjé: Nāgārjuna Meant What He Said

Of the Tibetan responses to Daktsang's presentation of Madhyamaka, the most radically antinomian was composed by Wangchuk Dorjé, who portrays Daktsang as one of the very few Tibetans who correctly understood Nāgārjuna and Candrakīrti. In the Karmapa's understanding, Madhyamaka properly understood is not a philosophical system—it rejects all attempts to create conceptual frameworks and eschews affirmations of any kind.

> That anything constitutes the Prāsaṅgikas' own system would entail the truth of any random idea. That anything is a proven entity (*gzhi grub*; Skt. *vastu*) would entail the truth of any random idea. To be conventionally existent precludes being ultimately nonexistent.[28]

Wangchuk Dorjé considers a shocked response from an unnamed opponent: "You can't be serious. Would you agree that Candrakīrti is a Mādhyamika?" Wangchuk Dorjé replies that indeed he makes no such assertion. "Is *Entry into to the Middle Way* (*Madhyamakāvatāra*) a Madhyamaka work?" The Karmapa acknowledges that many people say such things, but Mādhyamikas will only agree with the fact that an assertion has been made.[29] "Does this agreement constitute a thesis?" No, it merely describes what the Mādhyamika observes, but does not entail any commitment either way regarding the provenance of the treatise or the affiliation of its author. The true Prāsaṅgika (as opposed to people like Tsongkhapa who claim to follow Nāgārjuna but fail to understand the implications of his thought) is a thoroughgoing skeptic who applies the logic of emptiness to all philosophical claims, deconstructing them without feeling any need to put forward counterproposals.

> To be a proven entity entails freedom from fabrications. Does it also entail that such a thing is free from fabrications? Even if we use the copula "is," we would only commit to an extreme position involving fabrications if we were to do so with conviction; if we were thereby unequivocally to decide—in terms of our own system—between something's existing or not, or its being this or that; or if we were to make a statement about a particular extreme's existence, or about its being this or that. Simply saying that something is free from fabrications, however, does not amount to accepting a claim. [30]

Wangchuk Dorjé effectively jettisons Daktsang's framework of three contexts. Prāsaṅgikas make no assertions *in any context*. The fact that some people propound statements regarding what they refer to as Madhyamaka does not entail that such a thing exists. This, he claims, is Candrakīrti's intent, and Wangchuk Dorjé reads him as a thoroughgoing skeptic. Wangchuk Dorjé's presentation of Madhyamaka relies entirely on the perspective of ultimate analysis, which is also the perspective of noble beings, including buddhas. Noble beings operate within a realm in which all duality has been eliminated, along with tendencies to view "reality" in terms of conventional truths. Epistemic warrants and the notion of a "Madhyamaka view" only appear to be valid within the realm of the conventional world of truth and falsity; such notions have no traction for those who perceive reality as it is.

Wangchuk Dorjé's defense of Daktsang and his expansion of the critique of Tsongkhapa should be contextualized within the political situation in which he operated. His order, the Karma Kagyü, had been embroiled in armed conflicts with the Gelukpas for almost two centuries, and his position as the most prominent figure in the order meant that his work constituted a direct challenge to some of the fundamental principles of Tsongkhapa's system, which had become the state ideology of the Ganden Podrang (dGa' ldan pho brang), the government of the Dalai Lamas.

## 6. Purchok's Reformulation of Tsongkhapa's Approach

Purchok Ngawang Jampa's *Diamond Slivers: A Rejoinder to Taktsang the Translator*[31] is the third Geluk response to Daktsang's critique of Tsongkhapa, and it incorporates elements of the rebuttals of his Geluk predecessors Losang Chökyi Gyeltsen and Jamyang Shepa. Like them, Purchok adopts a format that is modeled on the system of dialectical

debate favored by Gelukpas. He also emulates them in attributing to Daktsang positions he either does not assert or explicitly rejects, and the text is replete with the sorts of hyperbolic attacks debaters commonly employ to rattle opponents. Much of it reads like an extended debate in which Purchok flings a series of unwanted consequences at Daktsang while the latter stands dumfounded, unable to muster an effective response. Like Losang Chökyi Gyeltsen and Jamyang Shepa, Purchok refuses to take seriously Daktsang's assertion that he is adopting a strictly Prāsaṅgika *reductio* approach, merely pointing out inconsistencies in Tsongkhapa's presentation without advancing any tenets of his own.

Near the beginning of *Diamond Slivers*, Purchok contends that there *is* a Madhyamaka view, and it is founded on deep realization of the true nature of reality. Tsongkhapa's system is the supreme articulation of Buddhism; he was an emanation of Mañjuśrī, and so there is no possibility of any other version of Buddhism approaching the nuanced and profound presentation of Madhyamaka found in the Master's works. For any rational person encountering Tsongkhapa's treatises, the only appropriate response is an attitude of reverence:

> A refutation of Tsongkhapa is out of the question for any thinking person; an independent-minded, careful, and intelligent scholar who studies his works in detail can only reasonably bow to them with folded hands, hairs of faith standing on end! Any attempt at refutation would only consist of redundant, unsound, or fallacious arguments. [32]

Daktsang, however, vastly overestimated both his own intelligence and his meditative attainments: "Due to misplaced confidence in the supremacy of his views, Daktsang . . . came to regard epistemologically warranted conventions as inimical to the Prāsaṅgika approach."[33] Contrary to his scholarly pretensions, Daktsang was a novice meditator, and because of this was incapable of reconciling the extremely subtle object of negation (viz., the objective existence of phenomena) with conventional functioning within his philosophical system. Thus Daktsang conflated what is accepted in the world based on conventional epistemic warrants with the subtle object of negation. Purchok asserts that everyone, even skilled philosophers, initially misidentifies the subtle object of negation, and the only way to overcome this tendency is to comprehend the Madhyamaka view through introspective meditation. Even Tsongkhapa had this flaw early in his life, before full awareness dawned in his consciousness. Fortunately for him, he had an omniscient tutor—something Daktsang lacked, which meant that the latter had to rely on his own limited intellectual resources.

Purchok recounts a well-known visionary experience in which Buddhapālita gave Tsongkhapa a copy of his commentary on *Fundamental Verses on the Middle Way* (Nāgārjuna n.d.), and after examining the text, Tsongkhapa

> unerringly developed within his mind the highest view of the Prāsaṅgika approach— the view that appearance never contradicts emptiness and emptiness never contradicts appearance—and that, moreover, appearance and emptiness never contradict each other. [34]

Tsongkhapa thus came to understand, both through intellectual investigation and meditative training, the unity of appearances and emptiness and how this is established through epistemic warrants in the Prāsaṅgika system. He realized that it is not contradictory for phenomena to be empty of the intrinsic existence attributed to them by obscured minds and still be able to perform functions—or for people to be able to arrive at verifiable knowledge. Things function as part of a universal matrix of interdependent causality, and there is no foundational standpoint on which one might base one's epistemology. Nonetheless, the operations of things can be discerned by perception and other epistemic instruments, and the ultimate truth can be grasped through ultimate analysis.

Purchok develops a reading of Nāgārjuna and Candrakīrti according to which non-foundational epistemic instruments can yield reliable knowledge in a conventional context. Things like causes and effects or agents and actions exist contingently; they are mere

appearances and labels whose specifications are dependent on an interconnected web of conventional meaning, but they lack any sort of objective existence. Nonetheless, it is still possible (and in fact necessary) for beings operating on this level to make sense of their surroundings and to employ epistemic instruments in a way that can produce reliable knowledge.

This is, however, contingent on future data. Purchok sets out a fallibilist version of Madhyamaka according to which people make use of perception, inference, testimony, analogy, and other instruments, noting regularities of cause and effect and what sort of epistemic practices most often produce successful pragmatic outcomes, as well as those that have consistently negative or counterproductive results.

Purchok's position is broadly coherentist, and he denies that the use of epistemic instruments entails foundationalism. These instruments are merely transactional, and they operate in such a way that they mutually support each other, like a bundle of sticks propped up against each other. If one is removed, the whole edifice crumbles, but as long as they buttress each other they can perform functions. All epistemic instruments rely on the others within a mutually reinforcing system of perception and justification, but this does not mean that it is untrustworthy, at least on the conventional level. Independent truthmakers bolster each other, and the structure, while lacking any underlying foundation, still functions well enough for conventional purposes.

Like any coherentist view, Purchok's vision of Madhyamaka raises the specter of relativism: if a general agreement regarding the outcomes of applying epistemic warrants is all we have, how many people are required to constitute a consensus? This is precisely the issue Daktsang identifies as a flaw in Tsongkhapa's system, and Purchok does not really address it in a satisfactory way. He considers the often-stated situation of various types of beings perceiving flowing water: humans see it as a river that they can use for drinking or swimming; gods (*deva*) perceive it as ambrosia; and hungry spirits (*preta*) are confronted with disgusting pus and blood. Purchok formulates this with a sort of epistemic perspectivalism: each type of being has a particular perceptual apparatus, and this yields knowledge that is valid within its purview. Hungry spirits receive sensory input that is valid for optimally functioning *preta* perception. Humans should see water, and those who do so know it in the way it exists conventionally. If someone has a question, she can consult others or seek expert opinion. Consensus will often be sufficient to warrant conclusions, but in some cases more rigorous measures may be required. If I see falling hairs, for example, and I discuss this with a group of people composed of some with myodesopsia and others with unimpaired vision and receive mixed responses, I may decide to have my eyes tested by an ophthalmologist.

Perception, according to Purchok, is constructive and not merely passive. All beings except buddhas are mistaken about the final mode of existence of the things they experience, but this is no reason to abandon knowledge acquisition. Purchok adds that proper and repeatable functioning is possible within the conventional realm. His epistemology is descriptive, rather than prescriptive: he details how epistemic functioning works for various sorts of beings but adds that there is no foundational standpoint within conventional reality that might ground any one of these.

Purchok addresses the question of the difference between mistaken perceptions like falling hairs and things perceived in dreams—which appear as real and can provoke real emotions and even physical movements—and contrasts them with the phenomena of the conventional world. We cannot ride a dream elephant or spend the wealth we acquire in a dream, nor can we brush the hairs of myodesopsia. Such pragmatic tests can be employed broadly within the matrix of the dependently arisen phenomena that constitute our world, as well as for the epistemic instruments we deploy to make sense of it. Al-though perceptions arise in dependence on their perceivers, and individuals within a class of beings will have diverging impressions and interpretations, this need not lead to solipsism or despair regarding the possibility of agreement, at least provisionally and conventionally. Waking states trump dreams, and unimpaired eyesight trumps myodesopsia.

> Until a person has awoken from ordinary sleep, the objects, sense faculties, and sensory cognitions contaminated by sleep exist; when a person wakes up, these three no longer exist. Similarly, so long as one has not awoken from the sleep of ignorance, the objects, faculties, and cognitions contaminated by ignorance exist; when, however, one eliminates one's predispositions for ignorance from their roots ... [it is not the case that] objects, faculties, and cognitions are utterly nonexistent at the stage of awakening. [35]

It is important to note what Purchok is not saying. Unlike Daktsang, he does not believe that valid conventional perception is undermined by a noble being's cognition of ultimate truth; each operates within its own sphere of authority. Buddhas and ordinary beings participate in the same world, albeit from differing perspectives and with differences of relative skillfulness in their actions and their outcomes. When someone attains buddhahood, her earlier mode of perception is sublated by awakened cognition, but the mundane world continues to function for ordinary beings, and they can arrive at repeatable and reliable knowledge of their surroundings.

None of this requires foundations. The test of reliability is whether or not an appearance accords with communally shared conventions and mutually constructed epistemic warrants. Buddhas who have transcended mundane reality can still operate effectively within this domain because they retain memories of how they employed epistemic instruments in past lives, and this serves to bridge the conceptual gap between the error-inflected minds of those they work to save and their own omniscient consciousnesses.

In this coherentist reading of Madhyamaka, ideas and epistemic practices that produce reliable and verifiable results will tend to persist, while those that do not will gradually be abandoned. Most people once believed that the sun revolves around the earth and that whales are fish, but subsequent evidence disproved those notions. The process of challenging and overturning accepted practices is often slow and contentious, but in Purchok's view it works well enough to allow for confidence and the possibility of (at least provisional) knowledge on the conventional level.

Purchok accuses Daktsang of mistakenly limiting the purview of buddhas' omniscience: if they only access ultimate truth, that would entail that they are unable to perceive half of reality, viz., the conventional—and this would mean that they would not truly be omniscient and that they would be ineffective in their efforts to bring sentient beings to liberation. Buddhas would be incapable of comprehending the thoughts of ordinary beings, and so they would inhabit a realm utterly dissociated from that of trainees. Purchok rejects this characterization and explains that like someone who helps a friend recover from a frightening dream, it is not necessary that buddhas experience exactly what their students do. The counselor can relate to another's dream because she has had similar experiences, and buddhas—who have unerring memories of their past lives—can access this information and so can, in effect, enter into the cognitive worlds of beings who operate within mundane convention.[36] Thus Purchok seeks to rescue the Madhyamaka project from the sort of nihilism he attributes to Daktsang and to preserve the account of valid epistemic warrant within the conventional realm that is at the core of Tsongkhapa's project.

## 7. Conclusions

Each of the authors we have examined draw out implications of statements in the works of Nāgārjuna and Candrakīrti that lend themselves to divergent interpretations. Depending on which passages one highlights and what sort of philosophical agenda one is pursuing, it is possible to read the Indian Mādhyamikas as antirealist nihilists; as propounding a transcendentalist view according to which only the buddha-perspective is valid; as a form of coherentism based on mutually agreed upon epistemic instruments that support each other but whose outcomes are subject to the inherent fallibility of our senses and consciousnesses; or as a perspectivalism that interprets validity in relation to particular sorts of beings, each operating within a closed system of perception and interpretation. As we have seen, Tibetan exegetes from different traditions arrived at each of these conclusions

in their readings of their Indian forbears, and the work of philosophical analysis continues today in Tibetan intellectual circles. The treatises of Nāgārjuna and Candrakīrti continue to be widely regarded as authoritative, but exactly what they intended is still very much open to debate.

**Funding:** Funding for this research was provided by an Australian Research Council Discovery grant (DP160100947).

**Institutional Review Board Statement:** Not applicable.

**Informed Consent Statement:** Not applicable.

**Data Availability Statement:** Not applicable.

**Conflicts of Interest:** The author declares no conflict of interest.

## Notes

1  There is a great deal of divergence in historical sources that describe this event, and a number of scholars have concluded that it probably did not actually occur, at least as a single winner-take-all contest; see Gómez (1983).

2  See Pasang Wangdu and Sørensen (2001), pp. 20–21.

3  Jacob Dalton (2014) provides a good overview of the points of contention. Sam van Schaik (2008, 2015) discusses documents attributed to Moheyan and his Chinese followers, as well as Tibetan works relevant to the debate, and develops a far more nuanced picture of Moheyan's thought than that found in traditional Tibetan sources.

4  This refers to an earlier passage in which an unidentified opponent accuses Nāgārjuna of self-contradiction because he proclaims that he has no thesis—but this claim itself constitutes a thesis.

5  Nāgārjuna (n.d.), *Reply to Objections* (*Vigraha-vyāvartanī*; Tib. *rTsod pa bzlog pa'i tshig le'ur byas pa*), GRETIL e-text: http://gretil. sub.uni-goettingen.de/gretil/1_sanskr/6_sastra/3_phil/buddh/nagyskr.txt (accessed on 5 October 2021). sDe dge #3828, bsTan 'gyur, dBu ma, vol. *tsa*: 28ab (vv. 29–30).

6  See, for example, Tillemans (2016), pp. 1–84 and Garfield (2011).

7  Candrakīrti discusses three etymologies for this term: (1) universal obscuration (*samantād varaṇam*), a comprehensive misunderstanding (*ajñāna*) that hides the nature of objects from the perceptions of sentient beings; (2) mutually coming together (*paraspara-saṃbhavana*), which refers to how phenomena come into being through "mutually supporting each other" (*anyonya-samāśrayeṇa*); and (3) accepted worldly discourse (*saṃketo loka-vyavahāraḥ*), the conventions practiced within epistemic and linguistic communities, which are based on accepted custom (*Clear Words*, Vaidya ed., Candrakīrti 1960, ch. 24: 214.8).

8  Candrakīrti (n.d.), *Commentary on Four Hundred Verses*: 197b.

9  Batsab Nyima Drakpa (2006), 49b.

10  Chaba Chökyi Sengé (1999), p. 66.

11  Mapja Jangchup Tsöndrü (2006): 27b–d (746).

12  Ibid., p. 29. For a detailed discussion of how Tibetans characterized the relations between Prāsaṅgika and Svātantrika Madhyamaka, see Dreyfus and McClintock (2003).

13  Daktsang (2007).

14  Ibid., p. 274.

15  Ibid., p. 273.

16  (Candrakīrti (n.d.), *Entry into the Middle Way*: 6.25, 6.81cd.

17  Daktsang (2007), p. 294.

18  Ibid., p. 307.

19  This refers to the Perfection of Wisdom (Prajñāpāramitā) discourses.

20  The four extremes are: is, is not, both is and is not, neither is nor is not.

21  Daktsang (2007), pp. 294–98.

22  See Smith (2001), p. 244.

23  Gyeltsen (1973). This work is discussed by Cabezón (1995).

24  Jamyang Shepa (1999). This is translated and analyzed by Jeffrey Hopkins (2003).

25  Hopkins (2003): 17 remarks on the text's "nasty" tone, which he characterizes as "at first shocking, then boring, and finally counter-productive."

26  Daktsang (n.d.), *A Brief Homage Regarding the Character of the Precious Master Losang Drakpa.* As Roger Jackson (Sopa 2009, 457 n. 1092) notes, "this verse is known only through its citations by Geluk authors." Gelukpas, however, accepted it as authentic. In

*Crystal Mirror of Good Explanations that Shows the Sources and Assertions of All Philosophical Systems*, Losang Chökyi Nyima (1963) (Thu'u bkwan bLo bzang chos kyi nyi ma, 1737–1802; Ibid., p. 262) comments that Daktsang "had his mountain of pride cast down, and attained the faith of a Dharma follower with regard to Tsongkhapa and praised him sincerely." Tügen later returns to a discussion of the verses and states that after publishing his misguided polemical work Daktsang devoted himself to study of Tsongkhapa's treatises and realized that they were perfect in every respect and accurately captured the true intent of the Buddha and Indian Buddhist luminaries.

27 See Ruegg (1963), pp. 89–90; Stearns (1999), pp. 69–73; and Thurman (1982), pp. 243–245, which contains a translation of another example of this genre of Geluk propaganda entitled *In Praise of the Incomparable Tsongkhapa* (*mNyam med tsong kha pa'i bstod pa*) attributed to Mikyö Dorjé (Mi bskyod rdo rje, 1507–1554), the 8th Karmapa. It describes Tsongkhapa as an incarnation of Mañjuśrī and threatens retribution by protector deities against anyone who critiques Tsongkhapa.

28 Wangchuk Dorjé (2005), p.1.

29 Ibid., p. 3.

30 Ibid., p. 1.

31 Purchok Ngawang Jampa (2012).

32 Ibid., p. 336.

33 Ibid., p. 336.

34 Ibid., p. 335.

35 Purchok Ngawang Jampa (2012): 343.

36 Ibid.: 347–348.

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
