# Peer review of "The Disputed Middle Ground: Tibetan Mādhyamikas on How to Interpret Nāgārjuna and Candrakīrti"

_religions, doi:10.3390/rel12110991_

Round 1
Reviewer 1 Report
Content: A very interesting, soundly argued, and original research that informs the reader about an ongoing debate using so far understudied/unstudied (by academics) commentarial sources. One could add some more context (and references) regarding the debate of Samye (Demieville etc.) or Madhyamaka in general (Ruegg's history and the like), but this is not a must.
The title mentions "Tibetan Madhyamikas..." As it is about a selected debate and specific authors, I suggest it be reflected in the title (alternatively, one may provide further examples of Madhyamaka discussion, but this would likely extend the scope of the paper).
Maybe two or three sentences about the historical contexts in which these masters operated could add to the paper's completeness.
Referencing: Those readers who know Tibetan would likely want to learn more about the Tibetan passages in their original for easier reference, but it might not be the policy of the Journal to keep Tibetan quotes in the footnotes.
Author Response
Point 1: I'm happy to add extra information on the Samye Debate, but it's not really relevant to the debate discussed in the article. When I was writing it, I tried to keep it as close to 8,000 words as possible (it's 8,400 words). I contacted the issue's editor, and he checked with editors at Religions, and the response was that 8,400 words is at the outer limit of what would be accepted. So adding material on the Samye Debate would require deleting material in the article that's more pertinent to the main topic.
Point 2: I don't see a problem with "Tibetan Mādhyamikas" in the title. There are thousands of Tibetan Mādhyamikas, so I can't imagine anyone thinking that a short article is a survey of all of them. The article examines the works of a few Tibetan Mādhyamikas who were part of the debate that is the focus of the article. There's no indication in the title that all Tibetan Mādhyamikas are being discussed.
Point 3: I agree that a discussion of the historical contexts of the authors would enhance the comprehensiveness of the article, but this would require several thousand words minimum. If the editor thinks this is desirable and is willing to allow more space, I'm happy to do this.
Point 4: adding Tibetan text would significantly add to the word count, and only Tibet specialists would benefit. The notes provide detailed information regarding where to find the relevant passages for anyone who wants to look them up.
Reviewer 2 Report
Excellent paper - though conclusion could be expanded.
Very compelling argument. No remarks
Author Response
The reviewer doesn't recommend any changes, so there's nothing to reply to.